# Design and Evaluation of Hybrid Composite Plates for Ballistic Protection: Experimental and Numerical Investigations

**DOI:** 10.3390/polym13091450

**Published:** 2021-04-30

**Authors:** Farah Alkhatib, Elsadig Mahdi, Aamir Dean

**Affiliations:** 1Department of Mechanical and Industrial Engineering, College of Engineering, Qatar University, Doha P.O. Box 2713, Qatar; farah.alkhateeb@live.com; 2School of Civil Engineering, College of Engineering, Sudan University of Science and Technology, Khartoum P.O. Box 72, Sudan; a.dean@sustech.edu

**Keywords:** ballistic protection, ballistic test, carbon fibers, Kevlar fibers, date palm fibers, FEM

## Abstract

In this paper, hybrid composite plates for ballistic protection were investigated experimentally and numerically, with a target to reduce the weight of currently used body armor inserts and, at the same time, satisfy the requirements of the National Institute of Justice’s (NIJ) ballistic protection standards. The current study has three phases to improve the ballistic plate’s energy absorption capability. The first phase is devoted to studying the effect of the material types, including three different fibers: carbon fiber, date palm fiber, and Kevlar fiber. The second phase is dedicated to studying the effect of hybridization within layers. The two previous phases’ results were analyzed to optimize the material based on the hybrid composite ballistic plate’s maximum energy absorption capability. The commercial finite element software package LS-DYNA was employed for numerical modeling and simulation. The hybrid composite ballistic plate could absorb more impact energy than the non-hybrid Kevlar plate with the same area density from the numerical simulation results. This study provides lighter-weight ballistic inserts with a high protection level, making movement easier for the wearer. The numerical results were verified by comparing results of a plate made of 40 layers of Kevlar with an actual ballistic test. The results indicated that the simulation results were conservative compared to the ballistic test.

## 1. Introduction

As stated by the Global Terrorism Index (GTI) [1], wars and terrorist attacks have increased in 30 years. During 2011 and 2014, armed assaults and explosions were the root cause of deaths by 400% compared to the previous decades. Therefore, the need for a high level of ballistic protection for military and defense sectors and personal protection for law enforcement and corrections officers has been a significant challenge for defense engineers against bullets and shrapnel [2,3,4]. Ballistic-resistant body armors played a considerable role in reducing injuries that might lead to disabilities and deaths in warfighting, counter-terrorism operations, and public security missions.

During Iraq’s war, 58% of wounds were in the eyes, legs, or hands, while 9% of wounds were in the torso [5]. According to statistical studies from international law enforcement agencies, ballistic-resistant body armors have saved more than 3000 police officers in the past years [6,7]. However, deaths and disabilities from penetrating projectiles are not the main problems in ballistic-resistant soft body armors; the massive amount of energy delivered to the chest tissues by a nonpenetrating projectile can cause fatal injuries, which are called the “Blunt trauma” [8,9,10,11].

The defense industry defines the term “personnel armors” as any protective clothing used to absorb impact energy from a fired gun or explosions, including ballistic shields, vests that cover the torso, helmets that cover the skull, masks, and goggles for face and eye protection [12,13]. The level of ballistic protection depends on the kinetic energy received from the projectiles and can be stopped by the armor [14]. The National Institute of Justice (NIJ) [15] provided fully described ballistic threats standards for each protection level and the projectile type.

Finding the optimum geometry and material for comfort, lightweight, and maximum energy absorption for body armor is challenging [3,16]. Most of the recent ballistic plate designs are based on material optimization; hybrid composite body armors are widely used for ballistic impacts. Liu et al. [17] have reduced ceramic and Kevlar layers’ thickness in ceramic ballistic plates for the same protection and lighter body armor. Alternatively, introducing small-scale energy absorption devices as a core in the armor plates and forming sandwich panels is a promising technique in the ballistic field [2,18,19,20,21,22,23].

Most of the previously studied composite ballistic plates under impact were conducted numerically [24,25,26,27]. This research performed an actual ballistic test on composite plates with different material sequences to evaluate their ballistic performance on the macro and micro levels. We introduced three different fibers: carbon fiber, Kevlar, and date palm natural fiber. Practical applications for Kevlar fibers, carbon fibers, and date palm fibers are ballistic protection products, aerospace/automotive applications, and indoor applications. We expected that the use of date palm fibers could enhance these composite plates’ ballistic properties, besides the economic and environmental advantages [28]. Conversely, results showed that introducing these natural fibers decreased the ballistic performance and increased its weight. If we used treated date palm fibers, the results would be different, with massive weight reduction.

## 2. Experimental Investigation

### 2.1. Geometrical Configuration and Sample Preparation

In this study, two geometries of plates, flat and curved, were designed according to small arms protective insert (SAPI) plate sizes. The dimensions of the plates were 10” × 12”, i.e., 250 mm × 300 mm.

Different sequences of three woven fabrics were used in fabricating the plates: Kevlar^®^, carbon fiber (Easy Composites, Staffordshire, UK), and Qatari date palm fiber; (Qatar University Farm, Doha, Qatar) 40 layers in total for all configurations. The flat and curved plates were fabricated using a vacuum infusion process (Figure 1). The resin was infused into the dry fabrics, depending on the pressure difference. During the fabrication processes, efforts and care were given to ensure ideal bonding between layers and excellent surface finishing. The curing process took 24 h under atmospheric conditions; then, the plates were transferred to an oven (80 °C) for 8 h. Table 1 summarizes the fabricated samples (three samples from each plate) with their characteristics.

Plates #1 and #2 had 40 layers of woven Kevlar. For the rest of the plates (i.e., plates #3–#7), carbon fibers were added to examine their effect on ballistic protection. Since carbon fibers have high strength-to-weight, using them in ballistic plates may reduce the plate’s weight while having a high level of ballistic protection. The material sequence used for plates #3 and #4 is (4 layers CFRP–6 layers KFRP) × 4. For plates #5 and #6, their design was inspired by the currently used ceramic faced/composite backed body armor plate. For the fabricated #5 and #6 plates, the ceramic layer was replaced by many CFRP layers to evaluate the carbon fiber efficiency against the tip of the bullet before penetrating the KFRP layers. We believe that the replacement of ceramic by CFRP layers will considerably reduce the plate’s weight. Finally, in Plate #7, Qatari date palm fibers (Figure 2d) were introduced inside the plate. Using natural fibers in body armors may have a moral and environmental impact. This plate’s layer configuration was 4 CFRP layers as the armor plate’s face and six layers of DPRP in-between the KFRP layers.

### 2.2. Ballistic Test

Ballistic testing is a way to check the resistance of body armor to penetration. It is a destructive test, but it does not measure the stresses on a sample or the energy absorbed by the sample; it only measures the acceptable number of partial and complete penetrations and the depth of the back-face signature (BFS) for partial penetration of the body armor, which should not exceed an acceptable limit. The way of evaluating a partial or complete penetration is by placing a clay backing material behind the armor plate; once the bullet passes into the clay, that means complete penetration occurred. Furthermore, if the bullet penetrates the armor plate partially, the bulge’s depth created on the clay will be measured and should not exceed 44 mm according to the NIJ ballistic resistance standard [15], a technical document that assigns the essential performance requirements of personal body armors. It has a detailed description of the testing procedures required to test and approve any personal body armor. The backing clay material is also called “Roma Plastilena,” this clay is twice the density of human tissue, and therefore it does not match a human’s specific gravity. For accurate BFS measuring, this clay is a plastic material that will not recover its shape elastically [29]. In this study, the fabricated flat and curved plates’ ballistic behaviors were carried out by shooting test (three shoots per plate) in an open space with a 15 m distance from the body armor plates. The 9 mm FMJ RN bullet was used for all the samples (Figure 3), with an initial impact speed of 398 m/s ± 9.1 m/s [15]. Ultimately, BFS was measured for each body armor plate. One of the limitations of this work is that the ballistic impact history was not recorded, and this will be done in future studies using a high-speed camera.

## 3. Numerical Investigation

### 3.1. Computational Grid

The flat body armor plate and the bullet were drawn using SOLIDWORKS software with detailed dimensions, as shown in Figure 1a and Figure 4. The finite element (FE) models were developed and built using the pre-processor LS-PrePost (V4.6, Livermore Software Technology Corporation LSTC, Livermore, CA, USA). The flat plate mesh was biased with different element sizes due to the symmetry of the plate. The instantaneous heat generated due to the impact and fracture was not considered in this study.

The generated FE models were analyzed using the commercial FE software LS-DYNA (mmps R8.1.1, Livermore Software Technology Corporation LSTC, Livermore, CA, USA). The solved models were analyzed using the post-processor from LS-PrePost. In this study, we modeled the flat non-hybrid [KFRP]_40_ plate (#1) and the flat hybrid [CFRP_10_/KFRP_30_] plate (#5) by impacting them with a 9 mm FMJ RN bullet (Figure 4). Table 2 summarizes the number of elements and nodes of the modeled composite plates and the projectile.

### 3.2. Material Modeling & Failure Criteria

The following material models were used in this FE model:MAT_058 (MAT_LAMINATED_COMPOSITE_FABRIC) was used to model the composite woven fabric layers. The KFRP and CFRP were represented as shell elements in this FE model. This material’s failure criteria are based on Matzenmiller’s damage mechanics model, which can model the damages independently in the principal axis direction of any orthotropic material [30]. This makes it suitable for fabric composites. The elastic orthotropic material parameters for the longitudinal (x-direction), transverse (y-direction), and normal (z-direction) directions were defined to simulate the composite failure and predict the delamination accurately. Table 3 summarizes the defined material properties in MAT_058 [31,32].MAT_010 (MAT_ELASTIC_PLASTIC_HYDRO) was used to model the bullet used in real shooting tests, as this material model is suitable for hydrodynamic materials.

### 3.3. Interactions

The following contact conditions were used in this FE model:AUTOMATIC_SURFACE_TO_SURFACE_TIEBREAK was imposed to model the bonding between the composite layers (Kevlar and carbon fiber). In its formulation and during the loading, the damage of the material was a linear function of the distance between the two points which were initially in contact. When a critical opening was reached, the contact was being broken, and the bonded composite layers were converted into two separate surfaces with the regular surface-to-surface contact between them, see [23].AUTOMATIC_SURFACE_TO_SURFACE was defined between the bullet and the composite plate since this contact can deal with the deformations resulting from the bullet impact.ERODING_NODES_TO_SURFACE was used to simulate the erosion between the bullet and the composite layers in the body armor plates, as an erosion parameter is defined as a part of the material failure criteria in MAT_058 to delete any failed element from the simulation in order to save computational time, since the time-step is automatically adjusted, and to avoid random contact between failed elements.Between the composite layers, a tiebreak contact formulation was used. The tiebreak contact was used here as adhesive to bond the laminates. In its formulation and during the loading, the damage of the material was a linear function of the distance between the two points which were initially in contact. When the critical opening was reached, the contact was being broken, and the sub-laminates were converted into two different surfaces with the regular surface-to-surface contact between them, resulting in a strong displacement discontinuity.

### 3.4. Boundary Conditions

The body armor plates were rigidly constrained in x and y-axis directions, while the z-axis (the direction of the shot bullet) was not fixed. According to the NIJ standard, an initial velocity (398 m/s) was defined to model the impact velocity from the 9 mm FMJ RN bullet, similar to the actual test [15].

## 4. Results

In this section, the ballistic test’s experimental and numerical results are presented and discussed in detail. Using the back-face signature evaluation criterion, we found the new body armor plate’s optimum material sequence and geometry.

### 4.1. Experimental Results

Table 4 summarizes the results of the ballistic test of all tested plates. It was observed that plates #1 and #2 (i.e., flat and curved non-hybrid [KFRP]_40_ plates) and plates #5 and #6 (i.e., flat and curved hybrid [CFRP_10_/KFRP_30_] plates) succeeded in stopping the bullets in the three trials, and no full penetration occurred. Simultaneously, a full penetration was observed for plates #3 and #4 (i.e., flat and curved [CFRP_4_/KFRP_6_]_4_ plates). Although carbon fiber has an excellent strength-to-weight ratio, its ballistic properties are weak compared to Kevlar and introducing carbon fiber in-between the Kevlar layers decreased Kevlar ballistic properties against the bullet.

Figure 5 shows the deformed plates after the ballistic test. In Figure 5b, the bullet took a path inside the flat non-hybrid [KFRP]_40_ armor plate (plate #1) and was arrested between the Kevlar fiber layers. Similar behavior was observed for the curved non-hybrid [KFRP]_40_ armor plates (plate #2) for all three shots.

In contrast, plate #7 (i.e., flat [CFRP_4_/[DPRP_2_/KFRP_10_]_3_] plate) showed different ballistic properties in the three trials. The plate was observed to succeed in some shots where partial penetration was observed, or the bullet was stopped entirely between the layers (Figure 5i), or full penetration occurred (Figure 5h). It is worth mentioning that the involvement of natural fiber is exciting since it is low cost. The penetration and failure in stopping the bullet can be easily attributed to the random distribution of natural fibers and their critical length. Therefore, using natural fiber in body armor needs more investigation.

Additionally, we observed that the deformation of the bullet inside plates #5 and #6 (i.e., flat and curved hybrid [CFRP_10_/KFRP_30_] plates) was greater compared with the bullet’s deformation in plates #1 and #2 (i.e., flat and curved non-hybrid [KFRP]_40_ plates). It is also essential to mention that the flat plate’s ballistic behavior was found to be similar to the curved plate. However, curved body armor plates were more compatible with the human’s torso, which gives the soldier more comfort while wearing it.

### 4.2. Numerical Results

The shooting test of the flat non-hybrid [KFRP_40_] (plate #1) and the flat hybrid [CFRP_10_/KFRP_30_] (plate #5) plates were simulated using LS/DYNA finite element code, and the ballistic behavior is presented and discussed in this section. At first, a convergence study was conducted to ensure that refinement in the mesh chosen was adequate and sufficient. Three different meshes were generated to meet this objective. Following computation, it was found that the kinetic energy of the bullet for each of the three chosen meshes differs by less than one percent.

In Table 5, the numerical results are summarized and compared with the experimental ballistic test. For the flat non-hybrid [KFRP_40_] plate (plate #1), the bullet stopped at the 29th layer of KFRP, creating a back-face signature (BFS) of 22.5 mm. In the flat hybrid [CFRP_10_/KFRP_30_] plate (plate #5), the bullet stopped at layer number 37, creating 36.8 mm displacement (BFS). Both BFS values (i.e., 22.5 and 36.8 mm for plate #1 and plate #5, respectively) are acceptable according to NIJ standard [15], which defines BFS to 44 mm.

One of the common ways to evaluate body armor’s ballistic performance is the kinetic energy absorbed by the plate [33,34,35]. In this study, the 9 mm FMJ RN bullet has an approximate mass of 8 g and an impact velocity of 398 m/s [15]. Hence, the kinetic energy delivered to the plate is around 633 J (Equation (1)).
(1)$$KE=\frac{1}{2}m\left(v_{s}^{2}-v_{r}^{2}\right)$$
where *m* is the projectile’s mass, *v_s_*. is the projectile’s striking velocity, and *v_r_* is the projectile’s residual velocity.

Figure 6 shows bullet kinetic energy versus BFS for both simulated plates. It can be seen clearly that the BFS for plate #1 ([KFRP]_40_) is significantly less than plate #5 ([CFRP_10_/KFRP_30_]). The effect of CFRP layers had dominated kinetic energy until BFS of 10 mm, after which KFRP layers dominate. It was also computed that the CFRP layers caused the bullet tip to be highly deformed compared with the KFRP, as shown in Table 5.

## 5. Discussion

### 5.1. Macro-Energy Dissipation Mechanism of Armor Plates

#### 5.1.1. Fabric Deformation & Destruction

As the bullet hits the composite armor plate, the layers start to deform in a flexural manner. After fabric deformation, matrix cracking occurs to cause fiber debonding and breakage. At this point, the energy from the bullet is dissipated, and the composite layers start absorbing the kinetic energy by deforming the layers.

In plates #1 and #2 (i.e., flat and curved non-hybrid [KFRP]_40_), the first layers of KFRP faced the striking bullet, then the matrix cracking, fiber debonding, and breakage occurred. The bullet continued to deform and penetrate the KFRP until its kinetic energy was dissipated through the layers.

In plates #3 and #4 (i.e., flat and curved hybrid [CFRP_4_/KFRP_6_]_4_), the bullet bent the four-facing CFRP; these layers were deformed after the matrix cracking and fiber breakage. On the other hand, the subsequent six layers supported the armor plate from penetration coming from the bullet’s kinetic energy. Although these layers were trying to avoid penetration, the matrix cracking, fiber breakage, and deformation in fabric made them easy to penetrate. The exact mechanism occurred for the next CFRP and KFRP layers to end up with complete penetration.

In plates #5 and #6 (i.e., flat and curved hybrid [CFRP_10_/KFRP_30_]), the ten CFRP tends to bend then deform from the impact from the bullet, causing penetration due to matrix cracking and fiber breakage. The KFRP acted as support, same as what happened in plates #3 and #4, but the thirty layers were enough to arrest the bullet between them.

#### 5.1.2. Thermal Energy

Some of this bullet’s kinetic energy is transferred into thermal energy because of the friction between the bullet and fibers. A high thermal expansion coefficient (α) means more heat energy is absorbed by the material, which means more deformation between the composite layers. Although Kevlar and carbon fiber are resistant to very high temperatures, Kevlar’s destruction will be greater than for carbon fiber, as α for Kevlar is 3.5 times higher than carbon fiber.

#### 5.1.3. Wave Propagation

Moreover, some of the bullet’s kinetic energy is dissipated as two primary waves, transverse wave and longitudinal wave, depending on the material’s sound velocity at the point of impact (Figure 7) [36]. The longitudinal wave travels outward along the fiber axis, according to Equation (2), to cause stretching in fibers and make an in-plane movement. While the transverse wave deflects the fibers vertically to cause an out-of-plane movement, the transverse wave speed can be calculated from Equation (3).
(2)c=Eρ
(3)$$u=c\left(\sqrt{\varepsilon \left(1+\varepsilon \right)}-\varepsilon \right)$$
where *c* is the longitudinal wave speed (m/s), *E* is material Young’s modulus (GPa), *ρ* is the mass density of the material (kg/mm^3^), and *ε* is the fiber’s strain.

### 5.2. Micro-Energy Dissipation Mechanism of Armor Plates

Scanning electron microscopy (SEM) analysis was performed on the strike faces of the shot plates. SEM was carried out to examine and identify bullet kinetic energy’s energy dissipation mechanism through the micro-level ballistic plates. Figure 8 shows SEM images for strike faces of hybrid and non-hybrid body armor plates.

Figure 8a shows the micro failure mechanism and micro delamination for plate #1 (i.e., flat non-hybrid [KFRP_40_] plate). As the bullet strikes the face layers, matrix cracking was observed to initiate the kinetic energy dissipation mechanism. This mechanism was followed by fiber debonding and fiber breakage (macro-energy dissipation mechanism, see Section 5.1). The bullet progresses through the rest of the layers until the bullet arrests at BFS of 20 mm.

Figure 8b presents the SEM image for plate #3 (i.e., flat hybrid [CFRP_4_/KFRP_6_]_4_ plates), in which the first four CFRP layers are designed to face the strike, while the first six KFRP layers are designed to (1) support the CFRP layers, (2) prevent bending of CFRP layers, and (3) decelerate the bullet’s penetration. Bending was observed to dominate the first stage of the bullet’s strike, which delayed the bullet penetration. This bending response results in low energy dissipation. At the end of bending, the bullet penetrates the CFRP layers, and a small amount of energy is believed to be dissipated in the form of matrix cracking and delamination between the CFRP layers and KFRP layers. The rest of the kinetic energy was dissipated in the KFRP fiber breakage, and massive destruction on KFRP was observed. This failure mode results in the complete penetration of the plate. Existing CFRP layers between the KFRP increased the deformation throughout the entire plate thickness, which decreased the energy dissipation mechanism in the form of KFRP destruction.

The results from plate #3 inspired us to develop the new design by placing the CFRP layers at the strike face and the KFRP layers at the armor plate’s back face. Figure 8c shows an SEM image for plate #5 (i.e., flat hybrid [CFRP_10_/KFRP_30_] plate). As the bullet strikes the CFRP layers, small deformation of the CFRP layers was observed, while the KFRP supported and prevented the CFRP layers from experiencing more deformation. This small amount of deformation led to matrix cracking at the CFRP layers and accelerated the penetration of CFRP layers, resulting in high energy dissipation. As the bullet progressed, the KFRP layers started to absorb the bullet’s kinetic energy in destruction failure modes, such as matrix cracking, fiber debonding, fiber breakage, and delamination between layers that occurred at BFS of 26.2 mm, at which the bullet was arrested, and no complete penetration was observed.

Figure 8d presents the SEM image for plate #7 (i.e., flat [CFRP_4_/[DPRP_2_/KFRP_10_]_3_] plate). This material sequence has four CFRP layers at the plate’s strike face, penetrated and deformed due to matrix cracking and fiber breakage. The next two layers of the date palm fiber (DPRP) did not support the previous CFRP due to their weak ballistic properties. The DPRP faced massive matrix cracking, as these fibers were used without treatment. Thus, these fibers absorbed a small amount of epoxy in the fabrication process. The next six KFRP layers supported the DPRP, but the bullet’s kinetic energy was high enough to penetrate and deform the KFRP after the matrix cracks and Kevlar fiber breakage. The next two layers of DPRP did not minimize the bullet’s kinetic energy to cause penetration and deformation in the layers. This mechanism was repeated through the layers to cause complete penetration in the armor plate.

## 6. Conclusions

Based on the results and discussion of experimental and numerical programs, one can conclude the following:The ballistic behaviors of flat and curved ballistic plates are identical.The material stacking sequence significantly affects the hybrid composite plates’ energy dissipation mechanism and, consequently, the energy absorption capability.Plates #5 and #6, the flat and curved [CFRP_10_/KFRP_30_], where the material sequence was ten layers of carbon fiber and 30 layers of Kevlar, displayed the highest energy absorption capability and passed the actual ballistic shooting test.Plates #3 and #4 with material sequence [CFRP_4_/KFRP_6_]_4_ had low energy dissipation mechanism and did not pass the ballistic shooting test.Incorporating the untreated date palm natural fibers in the plate’s material sequence displayed good ballistic behavior, although it did not pass all three-trial ballistic tests. At the same time, the plate’s weight was more than the other tested plates. Hence, nanocomposites can be incorporated [37,38].Sophisticated non-local damage models based on the phase-field approach to fracture can be employed to improve the numerical prediction, see [39,40,41].

## Figures and Tables

**Figure 1 polymers-13-01450-f001:**
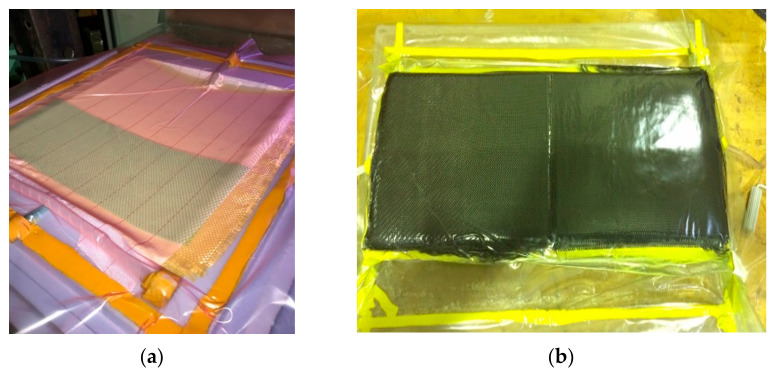
Vacuum infusion fabrication process: (**a**) back-side of the curved non-hybrid [KFRP]_40_ plates (plate #1), and (**b**) front-side of the curved hybrid [CFRP_10_/KFRP_30_] plate (plate #6).

**Figure 2 polymers-13-01450-f002:**
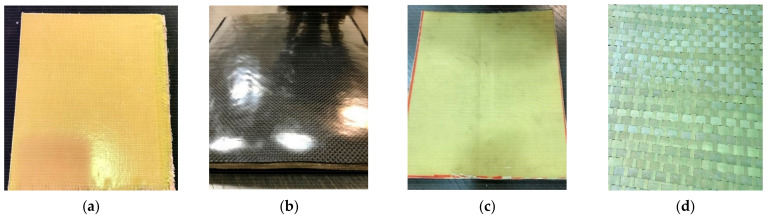
Examples of the fabricated body armor plates: (**a**) front-side of the flat non-hybrid [KFRP]_40_ plates, (**b**) front-side of the curved hybrid [CFRP_10_/KFRP_30_] plate, (**c**) back-side of the curved hybrid [CFRP_10_/KFRP_30_] plate, and (**d**) untreated Qatari date palm fibers.

**Figure 3 polymers-13-01450-f003:**
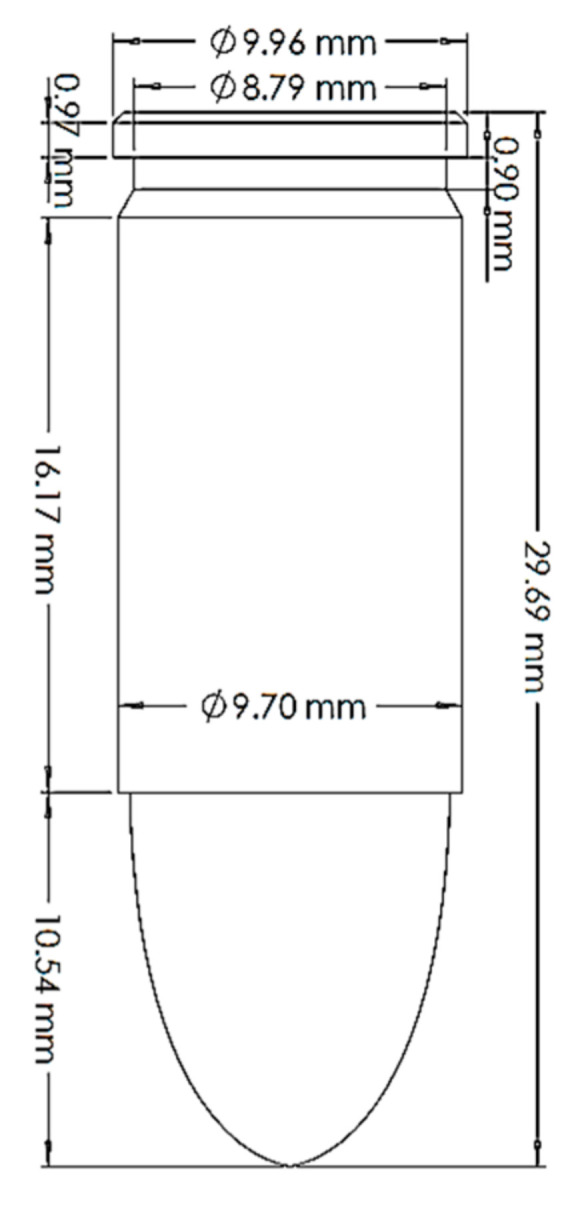
The geometry of the 9 mm FMJ RN bullet.

**Figure 4 polymers-13-01450-f004:**
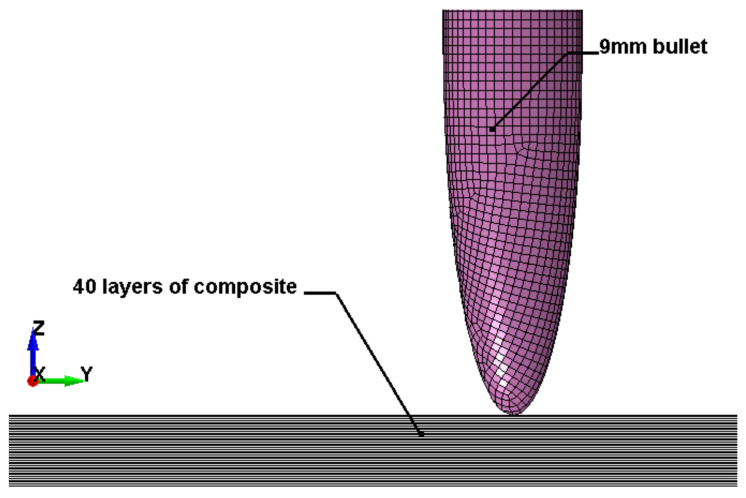
Section view of finite element model for the flat composite body armor plate.

**Figure 5 polymers-13-01450-f005:**
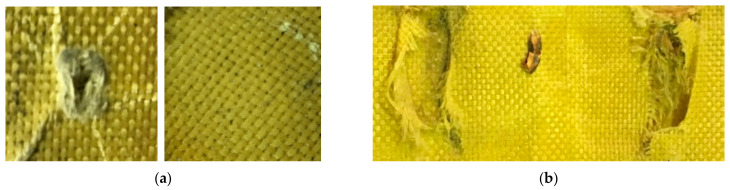
The deformed plates after the ballistic test: (**a**) strike face (left) and back face (right) of the flat non-hybrid [KFRP_40_] plate (#1), sample 1, shoot 1; (**b**) path of the stopped bullet between KFRP layers in the flat non-hybrid [KFRP]_40_ plate (#1); (**c**) strike face (left) and back face (right) of the curved non-hybrid [KFRP_40_] plate (#2), sample 2, shoot 1; (**d**) strike face (left) and back face (right) of the flat hybrid [CFRP_4_/KFRP_6_]_4_ plate (#3), sample 1, shoot 1; (**e**) strike face (left) and back face (right) of the curved hybrid [CFRP_4_/KFRP_6_]_4_ plate (#4), sample 3, shoot 1; (**f**) strike face (left) and back face (right) of the flat hybrid [CFRP_10_/KFRP_30_] plate (#5), sample 1, shoot 1; (**g**) strike face (left) and back face (right) of the curved hybrid [CFRP_10_/KFRP_30_] plate (#6), sample 2, shoot 1; (**h**) strike face (left) and back face (right) of the flat [CFRP_4_/[DPRP_2_/KFRP_10_]_3_] plate (#7), sample 3, shoot 1; (**i**) stopped bullet inside the [CFRP_4_/[DPRP_2_/KFRP_10_]_3_] plate.

**Figure 6 polymers-13-01450-f006:**
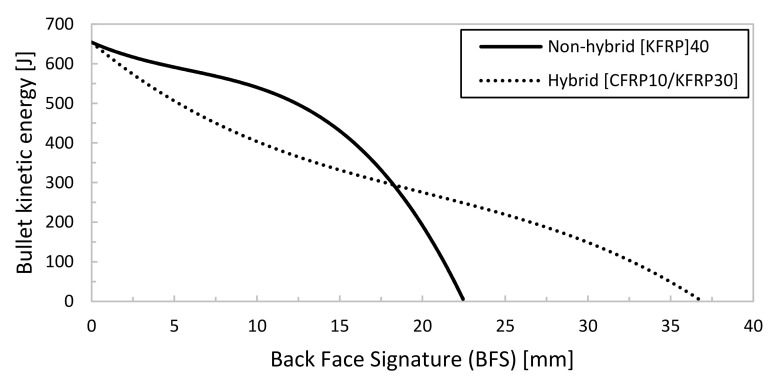
Bullet’s kinetic energy–BFS for non-hybrid [KFRP_40_] and hybrid [CFRP_10_/KFRP_30_] plates.

**Figure 7 polymers-13-01450-f007:**
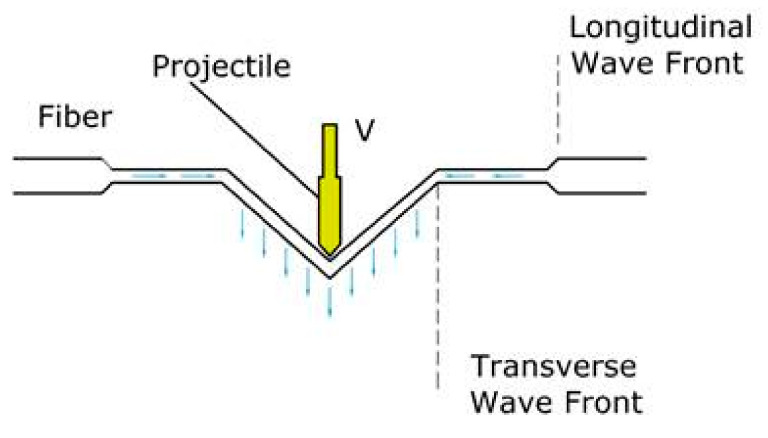
Transverse and longitudinal waves resulted in fiber from the bullet [35].

**Figure 8 polymers-13-01450-f008:**
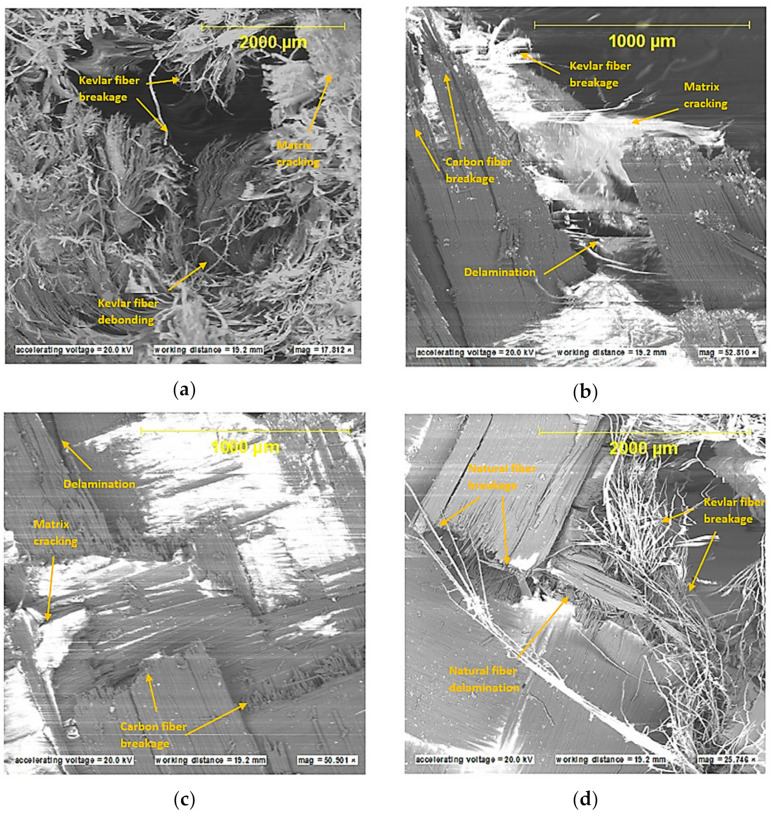
SEM images of strike face for the ballistic tested body armor plates: (**a**) flat non-hybrid KFRP plate (#1), sample 1, shoot 1; (**b**) flat hybrid [CFRP_4_/KFRP_6_]_4_ plate (#3), sample 1, shoot 1; (**c**) flat hybrid [CFRP_10_/KFRP_30_] (#5) plate, sample 1, shoot 1; (**d**) flat [CFRP_4_/[DPRP_2_/KFRP_10_]_3_] plate (#7), sample 3, shoot 1.

**Table 1 polymers-13-01450-t001:** Specifications of the fabricated plates, carbon fiber reinforced plastic (CFRP), Kevlar fiber reinforced plastic (KFRP), date palm reinforced plastic (DPRP).

Plate		Layers Configuration	Thickness [mm]	Weight [g]
#1	Flat non-hybrid [KFRP]_40_	[(KFRP)_40_]	10.1 ± 0.3	743.0 ± 18.7
#2	Curved non-hybrid [KFRP]_40_	[(KFRP)_40_]	9.9 ± 0.1	730.3 ± 4.5
#3	Flat hybrid [CFRP_4_/KFRP_6_]_4_	[(CFRP)_4_/(KFRP)_6/_(CFRP)_4_/(KFRP)_6/_(CFRP)_4_/(KFRP)_6/_(CFRP)_4_/(KFRP)_6_]	13.1 ± 0.3	842.0 ± 18.7
#4	Curved hybrid [CFRP_4_/KFRP_6_]_4_	[(CFRP)_4_/(KFRP)_6/_(CFRP)_4_/(KFRP)_6/_(CFRP)_4_/(KFRP)_6/_(CFRP)_4_/(KFRP)_6_]	12.6 ± 0.1	794.0 ± 5.3
#5	Flat hybrid [CFRP_10_/KFRP_30_]	[(CFRP)_10_/(KFRP)_30_]	11.1 ± 0.3	915.0 ± 8.0
#6	Curved hybrid [CFRP_10_/KFRP_30_]	[(CFRP)_10_/(KFRP)_30_]	11.1 ± 0.1	902.3 ± 8.1
#7	Flat [CFRP_4_/[DPRP_2_/KFRP_10_]_3_]	[(CFRP)_4_/(DPRP)_2_/(KFRP)_10_/(DPRP)_2_/(KFRP)_10_/(DPRP)_2_/(KFRP)_10_]	16.3 ± 0.2	1509.0 ± 14.6

**Table 2 polymers-13-01450-t002:** FEM details for the flat composite body armor plates.

Part	No. of Elements	No. of Nodes	Element Type
Composite layer	3600 × 40	3721 × 40	Shell
Bullet	34,503	36,564	Solid

**Table 3 polymers-13-01450-t003:** The material properties are defined in MAT_LAMINATED_COMPOSITE_FABRIC.

Mechanical Property	Woven CFRP	Woven KFRP
Mass density, ρ	1.6 × 10^−6^ kg/mm^3^	1.44 × 10^−6^ kg/mm^3^
Young’s Modulus, E_xx_ = E_yy_	175 GPa	18.5 GPa
Young’s Modulus, E_zz_	8.8 GPa	6 GPa
Shear modulus, G_xy_	5.5 GPa	1 GPa
Shear modulus, G_yz_ = G_zx_	2.5 GPa	5.43 GPa
Major Poisson’s ratio, ν_xy_	0.3	0.25
Minor Poisson’s ratio, ν_zx_ = ν_zy_	0.02545	0.33
Longitudinal compressive strength	850 MPa	190 MPa
Longitudinal tensile strength	1000 MPa	480 MPa
Transverse compressive strength	850 MPa	190 MPa
Transverse tensile strength	1000 MPa	480 MPa
In-plane Shear strength	670 MPa	50 MPa
Longitudinal compressive strain	0.8%	0.6%
Longitudinal tensile strain	0.85%	1.6%
Transverse compressive strain	0.8%	0.6%
Transverse tensile strain	0.85%	1.6%
In-plane shear strain	1.8%	1%

**Table 4 polymers-13-01450-t004:** Results of ballistic tested body armor plates (BFS: back-face signature).

Plate		Sample No.	BFS in mm		
			Shoot 1	Shoot 2	Shoot 3
**#1**	Flat non-hybrid [KFRP]_40_	1	20.0	24.3	22.5
		2	26.9	21.5	26.5
		3	23.5	22.3	33.5
**#2**	Curved non-hybrid [KFRP]_40_	1	27.2	22.5	29.8
		2	21.0	24.8	32.7
		3	26.8	31.1	35.0
**#3**	Flat hybrid [CFRP_4_/KFRP_6_]_4_	1	N/A *	36.1	42
		2	N/A *	N/A *	35.3
		3	N/A *	41.9	37.0
**#4**	Curved hybrid [CFRP_4_/KFRP_6_]_4_	1	39.0	N/A *	40.2
		2	N/A *	N/A *	43.2
		3	N/A *	N/A *	39.8
**#5**	Flat hybrid [CFRP_10_/KFRP_30_]	1	26.2	26	32.0
		2	21.3	22.5	23.4
		3	23.0	27.9	24.5
**#6**	Curved hybrid [CFRP_10_/KFRP_30_]	1	33.3	29.2	31.5
		2	24.2	26.7	21.0
		3	25.1	22.2	25.9
**#7**	Flat [CFRP_4_/[DPRP_2_/KFRP_10_]_3_]	1	38.2	40.5	N/A *
		2	41.0	N/A *	N/A *
		3	N/A *	N/A *	N/A *

N/A *: Non-applicable (i.e., complete penetration).

**Table 5 polymers-13-01450-t005:** Comparison between experimental and numerical strike faces (top views) and back-face signature (BFS) (front view) for the flat non-hybrid [KFRP_40_] and the flat hybrid [CFRP_10_/KFRP_30_] plates.

	Non-Hybrid [KFRP_40_] Armor Plate (Plate #1)	Hybrid [CFRP_10_/KFRP_30_] Armor Plate (Plate #5)
**Top view** 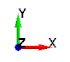	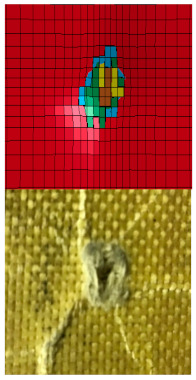	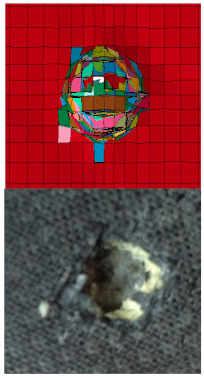
**Front view** 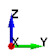	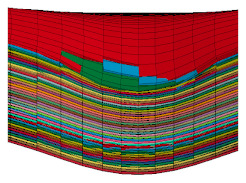 BFS: 22.5 mm (Experimental BFS: 20.0 mm)	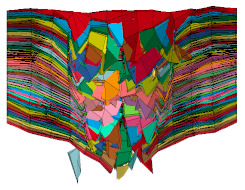 BFS: 36.8 mm (Experimental BFS: 26.2 mm)
**Bullet deformation after an impact** 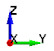	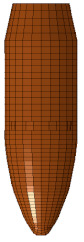	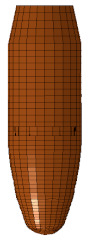	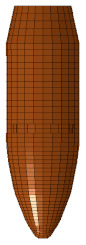	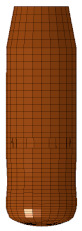
Undeformed bullet	Deformed bullet	Undeformed bullet	Deformed bullet

## Data Availability

The data presented in this study are available on request from the corresponding author.

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
