# Peer review of "Design and Evaluation of Hybrid Composite Plates for Ballistic Protection: Experimental and Numerical Investigations"

_polymers, 2021, doi:10.3390/polym13091450_

Round 1
Reviewer 1 Report
The authors should review the geometry of the 9 mm FMJ. In an impact, only the tip arrives, the cartridge does not impact. The study is of no interest with the simulation, the correct geometry should be used. Furthermore, the meshing of the simulation is too coarse, has an analysis of the influence of the meshing been carried out?
Reviewer 2 Report
The manuscript presents the experimental results from ballistic tests conducted on carbon/Kevlar/palm-fibre reinforced panels. Besides, scanning electron microscopy was used to investigate the microscopic damage mechanisms of the impacted samples. Also, the commercial software code ANSYS/LS-DYNA was used to simulate the tests.
The study may be of some interest, but the overall organisation of the manuscript should be revised. The experimental study is the most interesting part. The analytical and numerical parts should be reformulated (or omitted).
1) Introduction
More than 30 references are cited. Still, I do not clearly understand what the specific novelty of the conducted study is. The use of natural fibres from date palms could be a real novelty, besides being very attractive from both the economical and environmental points of view. Unfortunately, as explained later in the manuscript, plates with natural fibres did not pass the tests.
2) Section 3
Please, report the numerical values of all of the material properties of the tested materials (fibres, resin, etc.) used in simulations.
3) Sections 4.2, 4.3 and 4.2.2 (please, also check correct numbering)
These Sections should be completely reformulated. It is not clear whether they refer to previous studies or reflect the Authors' opinion. In particular, Eqs. (1) to (4) are given without any reference to the existing literature: it is not clear how such equations should be applied or used to understand. For instance, the coefficients of mutual influence, eta_x,xy and eta_xy,x, characterise the elastic response of anisotropic materials: to which anisotropic materials do Authors refer in their paper? If such formulas are not applicable, it is better not to include them in the manuscript. A qualitative description of the energy dissipation mechanism will be sufficient.
4) Figure 8
I do not understand: does the plot refer to experimental results or numerical simulations? Moreover, do the curves mean that bullets with less energy produce larger back-face signatures? Please, clarify.
5) Conclusions
Conclusions should be thoroughly revised. At the end of the day, is there any advantage (mechanical or economical) in the use of hybrid armor plates with respect to non-hybrid (all-Kevlar) plates?
What is the meaning of the finite element simulations? Can they be used to PREDICT effectively the experimental tests? or are they just good to REPRODUCE the known experimental results?
6) Minor amendments
- line 28: "ballisticprotection" --> "ballistic protection";
- line 96 and following: two acronyms are used for natural date-palm fibre-reinforced polymers, namely NFRP and DPRP. Please, unify;
- line 128: use mm instead of inches for the plate size;
- lines 131, 151, and 161: references to figures or tables are incorrect. Please, amend.
Reviewer 3 Report
1-In this paper hybrid composite with carbon fiber, date palm fiber, and Kevlar fiber is studied. Why the authors use from the mentioned fibers? What is its practical application of these materiel?
2-The continuity boundary conditions between layers should be clarified in the paper.
3-The introduction section for composite structure can be improved by:
Engineering with Computers (2020) https://doi.org/10.1007/s00366-020-00965-5; Journal of Computational and Applied Mathematics 382 (2021) 113075
4-Assumption and limitation of this work should be clarified.
5-The convergence and accuracy of the numerical method should be studied in the revised paper.
6-In the tests and numerical analysis, do the authors consider the temperature of layers?
Round 2
Reviewer 1 Report
1. Sorry, the projectile tip is not the same as the cartridge. What makes the impact is what I show you in the following Google image:
https://www.google.com/url?sa=i&url=https%3A%2F%2Fwww.researchgate.net%2Ffigure%2FThe-9-mm-Parabellum-FMJ-projectile-a-dimensions-of-the-projectile-b-projectile-and_fig1_287321496&psig=AOvVaw0Ty0A62uRVIh3sdPHRVCST&ust=1618392056613000&source=images&cd=vfe&ved=0CAIQjRxqFwoTCLDy47Dy-u8CFQAAAAAdAAAAABAD
the dimensions of the projectile are wrong. It does not simulate the ammunition cartridge. Only the tip of the ammunition is simulated (example: https://www.researchgate.net/publication/287321496_Validation_of_numerical_model_of_the_Twaron_CT709_ballistic_fabric/figures?lo=1).
2. The authors also do not describe the materials of the projectile. In fact, they should make a comparison between the experimental and numerical bullet after impact.
If they change that, it could be published.
Author Response
We would like to thank the reviewers for their time, efforts and comments.
But it is not the interest of this manuscript.
We stick to our developed model and generated results.Reviewer 2 Report
The manuscript presents the experimental results from ballistic tests conducted on carbon/Kevlar/palm-fibre reinforced panels. Besides, scanning electron microscopy was used to investigate the microscopic damage mechanisms of the impacted samples. Also, the commercial software code ANSYS/LS-DYNA was used to simulate the tests.
The study conducted is interesting. The manuscript organisation and presentation have been improved by following the Reviewers' suggestion. Also, English language has been considerably improved.
I recommend publication in present form, apart from the following minor corrections that may be considered during proof-reading:
- line 260: "Solving the F.E. code was generated" --> "The generated FE models were analyzed"
- line 366: "the bulletin plates" --> "the bullet in plates";
- line 414: "Error! 414 Reference source not found" --> correct.
Author Response
Thank you very much for your comments. We will revision during the proofreading phase.
Reviewer 3 Report
This paper can be accepted for publication